# Costs of ribosomal RNA stabilization affect ribosome composition at maximum growth rate

Diana Széliová [1,3], Stefan Müller [2,3] & Jürgen Zanghellini [1,3 ✉]

Ribosomes are key to cellular self-fabrication and limit growth rate. While most enzymes are proteins, ribosomes consist of 1/3 protein and 2/3 ribonucleic acid (RNA) (in *E. coli*). Here, we develop a mechanistic model of a self-fabricating cell, validated across diverse growth conditions. Through resource balance analysis (RBA), we explore the variation in maximum growth rate with ribosome composition, assuming constant kinetic parameters. Our model highlights the importance of RNA instability. If we neglect it, RNA synthesis is always cheaper than protein synthesis, leading to an RNA-only ribosome at maximum growth rate. Upon accounting for RNA turnover, we find that a mixed ribosome composed of RNA and proteins maximizes growth rate. To account for RNA turnover, we explore two scenarios regarding the activity of RNases. In (a) degradation is proportional to RNA content. In (b) ribosomal proteins cooperatively mitigate RNA instability by protecting it from misfolding and subsequent degradation. In both cases, higher protein content elevates protein synthesis costs and simultaneously lowers RNA turnover expenses, resulting in mixed RNA-protein ribosomes. Only scenario (b) aligns qualitatively with experimental data across varied growth conditions. Our research provides fresh insights into ribosome biogenesis and evolution, paving the way for understanding protein-rich ribosomes in archaea and mitochondria.

[1] Department of Analytical Chemistry, University of Vienna, Vienna 1090, Austria. [2] Faculty of Mathematics, University of Vienna, Vienna 1090, Austria. [3] These authors contributed equally: Diana Széliová, Stefan Müller, Jürgen Zanghellini. ✉email: juergen.zanghellini@univie.ac.at

The ribosome is at the core of any (known) self-replicating organism. In a process called translation, ribosomes read the instructions from messenger ribonucleic acids (mRNAs) to synthesize the corresponding proteins, including ribosomal proteins (rPs). This autocatalytic nature of ribosomes ultimately limits the doubling time of a cell to the period it takes a ribosome to synthesize itself[1,2]. In *Escherichia coli* this would be 6 min, assuming that the ribosome consists of a 55-protein complex of approximately 7400 amino acids (AAs) that is translated at a speed of 21 AA/sec[3]. In fact, even in growth-optimized *E. coli*, that doubling limit remains far from being reached[4]. Nonetheless, it has been proposed that ribosomes, not only in *E. coli*, have been subjected to strong selective pressure to minimize their own duplication time in order to speed up the production of all other proteins[5]. With this principle in mind, Reuveni et al.[5] explain why ribosomes have many rPs of similar length.

Ribosomes are structures that have developed over time by adding ribosomal ribonucleic acid (rRNA) and rP around a central core[6]. This core is considered to be a leftover from ancient translation systems that evolved alongside the genetic code. Different types of ribosomes have evolved in bacteria, archaea, and eukaryotes, but their overall structures are similar within each kingdom[7]. For example, the mass of ribosomes in prokaryotes is made up of approximately 63% rRNA and 37% rPs[8,9], whereas eukaryotic ribosomes have an equal mass distribution of rRNA and rPs[5,10,11]. Thus, the question arises whether there is an evolutionary advantage in having such a high RNA content.

It has been suggested that the ribosome composition can be understood as a competition for resources between rRNA synthesis and rP synthesis[12,13]. In particular, Kostinski and Reuveni[12] derived two upper bounds on growth rate resulting from two autocatalytic loops, one for rP production, and one for RNA polymerase (RNAP) and rRNA production. By analyzing allocation data from *E. coli*, they concluded that maximum growth rate occurs at the current ribosome composition of 2/3 RNA and 1/3 protein. However, the specific processes that limit the two autocatalytic processes remained elusive.

Here we aim to provide a mechanistic understanding of these processes. We set up a small (coarse-grained) model of a self-replicating cell and perform resource balance analysis (RBA)[14,15]. In particular, we vary ribosome composition and ribosome allocations (fractions of ribosomes allocated to the synthesis of different proteins) and maximize growth rate. For simplicity, we assume a constant ribosome mass.

We focus on the primary role of ribosomal proteins, which is stabilizing rRNA (by preventing its degradation or misfolding). Ribosome protein content might also affect other parameters, such as translation rate. Proteins are generally better catalysts than RNA[16], but the ribosome's catalytic core is formed by rRNA[17] and operates at a relatively slow catalytic rate compared to typical enzymes. This suggests that there is little evolutionary pressure to increase the catalytic rate. Furthermore, ribosomes with the lowest protein content, like the *E. coli* ribosome, exhibit the highest translation rates[18–20]. Therefore, we do not consider the impact on translation rate in this study.

We find that the costs of stabilizing rRNA strongly influence the optimal ribosome composition. If we neglect rRNA turnover, our predictions suggest the presence of RNA-only ribosomes (in contrast to experimental evidence). Taking RNA degradation into account increases its biosynthesis costs, and maximum growth occurs for a mixed (RNA+protein) ribosome.

## Results

We introduce a (coarse-grained) mechanistic model of a self-fabricating cell and investigate optimal ribosome composition using RBA. That is, we maximize growth rate under several sets of constraints. We validate the model by predicting RNAP fluxes (rRNA synthesis fluxes), and RNA to protein ratios at different growth rates. Ultimately, we predict maximum growth rate at different ribosome compositions.

**A small model of a self-fabricating cell.** We consider the small (coarse-grained) model of a self-fabricating cell depicted in Fig. 1. The cell imports a carbon source (C) and has two types of metabolic enzymes, one synthesizing amino acids (AA) from the carbon source and the other one synthesizing nucleotides (NT) from the carbon source and amino acids. RNA polymerase (RNAP) uses nucleotides to form the ribosomal RNA (rRNA),

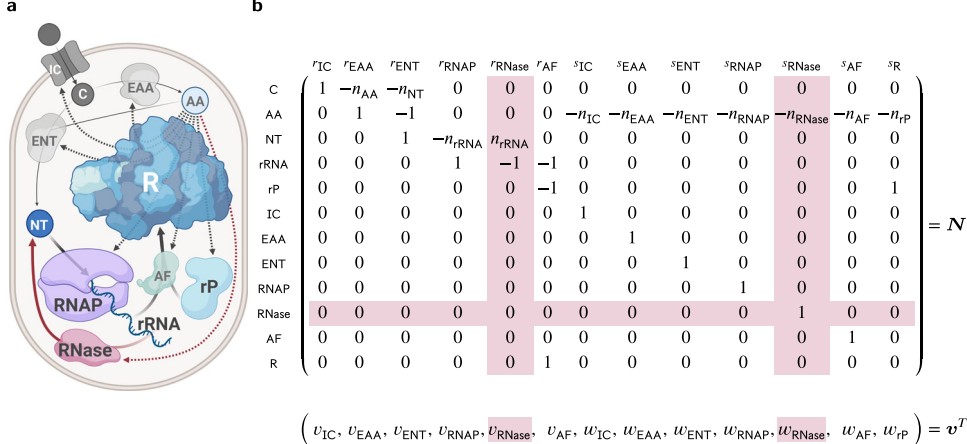

**Fig. 1 A small model of a self-fabricating cell. a** The cell imports a carbon source (C) and has two types of metabolic enzymes synthesizing amino acids (AA) from the carbon source and nucleotides (NT) from the carbon source and amino acids. The RNA polymerase (RNAP) uses nucleotides to form the ribosomal RNA (rRNA), and the ribosome (R) uses amino acids to synthesize the importer (IC), the metabolic enzymes (EAA, ENT), the ribosomal assembly factors (AF), and the ribosomal protein (rP). Finally, the assembly factors build the ribosome from ribosomal RNA and protein. The processes above constitute the base model. In the extended model, RNase degrades ribosomal RNA (and is synthesized by the ribosome). The additional processes are shown in red. Created with BioRender.com. **b** The resulting stoichiometric matrix and the corresponding flux vector. Here, $s$ is used for protein synthesis reactions (and $w$ for the corresponding fluxes), and $r$ is used for all other reactions (and $v$ for the corresponding fluxes). Additional columns and rows for the extended model are shown in red.

while the ribosome (R) uses amino acids to synthesize all proteins, including the importer (IC), the metabolic enzymes (EAA, ENT), the RNA polymerase and optionally a ribonuclease (RNase), the ribosomal assembly factors (AF), and the ribosomal proteins (rP). Finally, the assembly factors build the ribosome from ribosomal RNA and protein. In a base model, we neglect RNA degradation, whereas in an extended model we consider the enzyme (RNase) that breaks down RNA into nucleotides. We now provide a more formal definition of the two models.

Given the stoichiometric matrix $N$ and the vector of molar masses $\omega$, the dynamic model of cellular growth relates growth rate $\mu$, the vector of (metabolite, RNA, protein, and ribosome) concentrations $c$, and the vector of fluxes ($v$ for enzymatic reactions and $w$ for protein synthesis) according to

$$\frac{dc}{dt} = N \begin{pmatrix} v \\ w \end{pmatrix} - \mu c \quad \text{and} \quad \omega^T c = 1.$$

At steady state, growth rate $\mu$ and concentrations $c$ are determined by the fluxes $v$ and $w$,

$$N \begin{pmatrix} v \\ w \end{pmatrix} = \mu c \geq 0 \quad \text{and} \quad \mu = \omega^T N \begin{pmatrix} v \\ w \end{pmatrix}. \quad (1)$$

To take limited cellular resources into account, we consider capacity constraints for the enzymatic fluxes $v$, including transcription (and optionally RNA degradation),

$$v_i \leq k_i^{cat} c_i, \quad i \in \{IC, EAA, ENT, AF\}, \quad (2a)$$

$$n_{rRNA} v_{RNAP} \leq \bar{k}_{RNAP}^{el} c_{RNAP}, \quad (2b)$$

$$(n_{rRNA} v_{RNase} \leq k_{RNase}^{deg} c_{RNase}). \quad (2c)$$

Further, we consider the ribosome capacity constraint for the protein synthesis fluxes $w$,

$$\sum_{i \in Proteins} n_i w_i \leq \bar{k}_R^{el} c_R, \quad (2d)$$

$$Proteins = \{IC, EAA, ENT, RNAP, (RNase), AF, rP\}.$$

Here, $n_{rRNA}$ is the number of nucleotides in rRNA, $n_i$ is the number of amino acids in protein $i$, $k_i^{cat}$ is the corresponding enzyme turnover rate, and $\bar{k}_{RNAP}^{el} = k_{RNAP}^{el} f_{RNAP}^{act}$ and $\bar{k}_R^{el} = k_R^{el} f_R^{act}$ are the effective transcription and translation elongation rates, respectively. $f_{RNAP}^{act}$ is the fraction of actively transcribing RNAPs, and $f_R^{act}$ is the fraction of actively translating ribosomes. As mentioned above, RNase is synthesized optionally and hence put in brackets. By defining the ribosome allocations,

$$\phi_i^R = \frac{n_i w_i}{\bar{k}_R^{el} c_R}, \quad i \in Proteins, \quad (3)$$

that is, the fraction of ribosomes translating a certain protein $i$, constraint (2d) can be written as

$$\sum_{i \in Proteins} \phi_i^R \leq 1.$$

We refer to the model given by Equations (1) and (2a, b, d) as the base RBA model. Equations (1) and (2), including (2c), define the extended RBA model which additionally considers RNA degradation.

Throughout our study, we consider a fixed molar ribosome mass $\omega_R$, but variable rRNA and protein content,

$$\omega_R = n_{rRNA} \omega_{NT} + n_{rP} \omega_{AA},$$

and we study the influence of ribosome composition on the cell's maximum growth rate, under the constraints specified above.

Here, $\omega_{NT}$ and $\omega_{AA}$ are the molar masses of nucleotides and amino acids, respectively. For convenience, we define the ribosomal protein (mass) fraction

$$x_{rP} = n_{rP} \frac{\omega_{AA}}{\omega_R}, \quad (4)$$

and express $n_{rRNA}$ and $n_{rP}$ by $x_{rP}$,

$$n_{rRNA} = (1 - x_{rP}) \frac{\omega_R}{\omega_{NT}} \quad \text{and} \quad n_{rP} = x_{rP} \frac{\omega_R}{\omega_{AA}}.$$

In our analysis, we vary ribosomal protein fraction and maximize growth rate under given constraints. Modeling details can be found in section Methods/subsection Model, the stoichiometric, capacity, and (dry) mass constraints are summarized in Table 1, and the parameter values, variables and the units are given in Supplementary Table 1. We used parameters from E. coli grown in six different media. Three of them are rich media (Gly+AA, Glc+AA, LB) where amino acids (and nucleotides) are provided so cells only have to express the corresponding transporters instead of the synthesis pathways. In our model, the enzymes ENT and EAA represent lumped pathways for glycolysis and nucleotide/amino acid synthesis, and we only consider one type of transporter. Therefore, to model the changing nutrient quality of the different media (inspired by Scott et al.[21]), we assume that turnover numbers of EAA and ENT increase with growth rate.

**Base model recovers linear correlation of RNA to protein ratio with growth rate.** With parameters for E. coli in different media (listed in Supplementary Table 1) and the experimentally observed ribosome composition ($x_{rP} = 36\%$), the base model correctly recovers the well-known linear dependence of the RNA to protein ratio and growth rate[21], see Fig. 2a, but not the offset at zero growth rate, since our model does not contain any non-growth associated processes and we assume constant translation elongation rate $k_R^{el}$ as in Scott et al.[21]. At low growth rate, $k_R^{el}$ decreases, most likely because of the lower availability of the required substrates[20,22]. Interestingly, when we use variable $k_R^{el}$, we observe a nonzero offset (Supplementary Fig. 1).

To further test the model, we predict RNAP fluxes ($v_{RNAP}$) at various non-optimal growth rates in glucose minimal medium. In particular, we compute alternative solutions to the system of (in)equalities (1) and (2). (Technically, these solutions are elementary growth vectors as defined in Müller et al.[23]). We observe three lines (Fig. 2b). Two lines (in gray) correspond to solutions where either ribosomes or rRNA accumulate in excess of what is needed to support growth. In other words, constraints (2d) and (2b) (rows cap R and rRNA in Table 1) are not limiting. With increasing growth rate, the excess of rRNA and ribosome decreases, reaching zero at the maximum growth rate. The third line (in blue) corresponds to no accumulation of ribosomes or rRNA. In particular, the RNAP flux exactly matches the demand. At maximum growth rate, all lines converge to one optimal value.

For higher growth rates, experimental data are best fit by the line without accumulation of ribosomes or rRNA. In fact, the accumulation of free rRNA in a cell is biologically not realistic as it is bound by rPs already during transcription[24]. Furthermore, if rRNA is expressed in excess of rPs, it is rapidly degraded[25]. For the growth rates studied here (0.4–2.1 h$^{-1}$), the fraction of inactive ribosomes stays roughly constant at 15% to 20%[3,20,22]. In our model, we have already incorporated this fraction using the effective translation elongation rate ($\bar{k}_R^{el} = k_R^{el} f_R^{act}$). (However, below the growth rate of $\approx 0.5$ h$^{-1}$, the fraction of active ribosomes rapidly decreases[22]). Therefore, the disagreement

**Table 1 Constraints used in the extended and base models (with and without RNA degradation), see Fig. 1.**

| | $v_{IC}$ | $v_{EAA}$ | $v_{ENT}$ | $v_{RNAP}$ | $v_{RNase}$ | $v_{AF}$ | $w_{IC}$ | $w_{EAA}$ | $w_{ENT}$ | $w_{RNAP}$ | $w_{RNase}$ | $w_{AF}$ | $w_{rP}$ | sign | rhs |
|---|---|---|---|---|---|---|---|---|---|---|---|---|---|---|---|
| C | 1 | $-n_{AA}$ | $-n_{NT}$ | | | | | | | | | | | $=$ | 0 |
| AA | | 1 | $-1$ | | | | | | | | | | | $=$ | 0 |
| NT | | | $-1$ | $-n_{rRNA}$ | $n_{rRNA}$ | | | | | | | | | $=$ | 0 |
| rRNA | | | | $-1$ | $-1$ | $-1$ | | | | | | | | $\geq (=)$ | 0 |
| rP | $-\mu$ | | | | | $-1$ | | | | | | | $-n_{rP}$ | $\geq (=)$ | 0 |
| cap IC | $\frac{\mu}{k_{IC}^{cat}}$ | | | | | | $1$ | | | | | | | $\geq$ | 0 |
| cap EAA | | $\frac{\mu}{k_{EAA}^{cat}}$ | | | | | | $1$ | | | | | | $\geq$ | 0 |
| cap ENT | | | $\frac{\mu}{k_{ENT}^{cat}}$ | | | | | | $1$ | | | | | $\geq$ | 0 |
| cap RNAP | | | | $\frac{\mu}{k_{RNAP}^{el}}$ | | | | | | $1$ | | | | $\geq$ | 0 |
| cap RNase | | | | | $\frac{n_{rRNA}}{k_{RNase}^{deg}}$ | | | | | | $1$ | | | $\geq$ | 0 |
| cap AF | | | | | | $\frac{\mu}{k_R^{el}}$ | | | | | | $1$ | | $\geq$ | 0 |
| cap R | | | | | | | $k_{IC}^{cat}$ | $k_{EAA}^{cat}$ | $k_{ENT}^{cat}$ | $k_{RNAP}^{el}$ | $k_{RNase}^{el}$ | $\frac{k_{AF}^{cat}}{}$ | | $\geq$ | 0 |
| min deg | | | | | $\mu$ | $\frac{k_R^{el}}{\mu} k^{deg}(1-x_{rP})$ | | | | | | | | $\geq (=)$ | 0 |
| (dry) mass | $\omega_C$ | $-\mu n_{IC}$ | $-\mu n_{ENT}$ | $-\mu n_{rRNA}$ | $-\mu n_{rRNA}$ | $-\mu n_{AF}$ | $-\mu n_{IC}$ | $-\mu n_{EAA}$ | $-\mu n_{ENT}$ | $-\mu n_{RNAP}$ | $-\mu n_{RNase}$ | $-\mu n_{AF}$ | $-\mu n_{rP}$ | $\geq (=)$ | $\mu$ |

In particular, stoichiometric constraints (for the metabolites C, AA, NT, rRNA, rP), capacity constraints (for the catalysts IC, EAA, ENT, RNAP, RNase, AF, R), and the (dry) mass constraint. The column sign indicates an equality (=) or inequality (≥) constraint, and the column rhs specifies the right-hand side (a homogeneous or inhomogeneous constraint). The columns $v_{RNase}$, $w_{RNase}$ and the rows cap RNase and min deg are only present in the extended model.

between experimental and simulated data at lower growth rates is probably caused by neglecting other types of RNA. Indeed, RNAP allocation to the synthesis of different types of RNA changes with growth rate[3,20].

**Base model predicts maximal growth for RNA-only ribosomes.** We study the dependence of maximum growth rate on the ribosomal protein fraction using the base model described above. We find that, for realistic parameters from *E. coli* (Supplementary Table 1), rRNA synthesis is cheaper than protein synthesis for all tested growth conditions (see Fig. 3a). Thus, according to our base model, ribosomes should consist of rRNA only. Indeed, it has been suggested that higher growth rates could be achieved if ribosomes were to consist only of rRNA[5].

If we (hypothetically) adjust the parameters to make rRNA synthesis more expensive than protein synthesis (e.g. by decreasing $\bar{k}_{RNAP}^{el}$ or increasing $\bar{k}_R^{el}$), then maximum growth rate is achieved for a protein-only ribosome (Supplementary Fig. 2). By a symbolic analysis, we can rigorously prove that maximum growth rate is generically attained at an exclusive ribosome composition, either at $x_{rP} = 0\%$ or $x_{rP} = 100\%$, regardless of the parameters (see section Methods/subsection Symbolic analysis of growth rate maximization).

To conclude, RBA with standard capacity constraints does not explain mixed (RNA+ protein) ribosomes. Thus, additional constraints are needed.

**rRNA instability leads to maximal growth for mixed ribosomes.** As one potential explanation, we hypothesize that the different stabilities of rPs and rRNA affect the composition of the ribosome. While proteins are known to be highly stable[26], rRNA is susceptible to degradation by RNases, which are ubiquitous in cells[27]. Even at maximum growth, about 10% of rRNA is still degraded, and, thus, cannot be incorporated into the ribosome[27,28]. Furthermore, rRNA can easily misfold, rendering it inactive and prone to degradation[24,29].

To account for rRNA degradation, we introduce an RNase enzyme that breaks down rRNA into individual nucleotides (NT), via the reaction

$$r_{RNase}: \text{rRNA} \xrightarrow{\text{RNase}} n_{rRNA} \cdot \text{NT},$$

see Fig. 1. Since RNases are essential for quality control, we assume some minimum activity and add a minimum degradation rate,

$$v_{RNase} \geq k^{deg}(1-x_{rP}) c_R, \quad (5)$$

to the list of constraints (row min deg in Table 1). In the simplest case, this rate is directly proportional (with a constant $k^{deg}$) to the rRNA concentration. The latter is given by the fraction of rRNA in the ribosome concentration, since there is no free rRNA in the cell[3]. Additionally, $k^{deg} = k^{deg}(x_{rP})$ can be a (monotonically decreasing) function of $x_{rP}$,

$$k^{deg}(x_{rP}) = k_{max}^{deg}\left(1 - \frac{x_{rP}^n}{K^n + x_{rP}^n}\right) \quad (6)$$

modeling the cooperative protection of rRNA by proteins, where $n$ is Hill-factor and $K$ is half-saturation constant. As for the other enzymes, we add a capacity constraint for the RNase to account for its cost,

$$n_{rRNA} v_{RNase} \leq k_{RNase}^{deg} c_{RNase}, \quad (7)$$

where we use $k_{RNase}^{deg} = 88 \, \text{NT s}^{-1}$ of an enzyme called RNase R[30]. The base RBA model together with RNA degradation, RNase

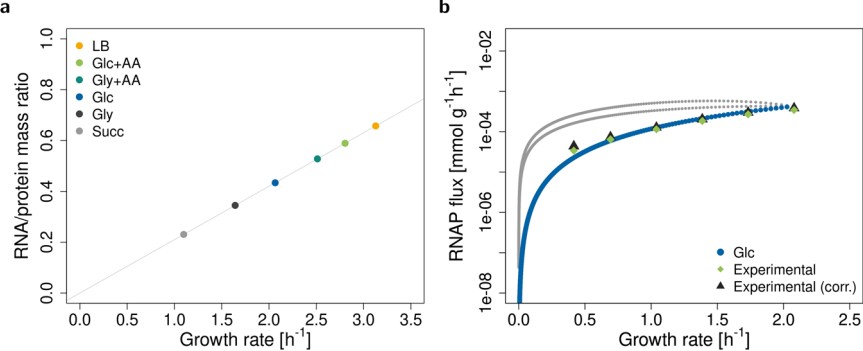

**Fig. 2 Validation of the base model. a** The model predicts a linear relationship between RNA to protein ratio and growth rate. The points represent the predicted maximum growth rates in six experimental conditions (Supplementary Table 1). The line is a linear fit. **b** Alternative RNAP fluxes at different non-optimal growth rates in glucose minimal medium. These are obtained by varying the growth rate step by step from zero to maximum and enumerating all solutions (elementary growth vectors as defined in Müller et al.[23]) for each growth rate. The grey and blue lines are the alternative solutions. The blue line corresponds to solutions, where rRNA and ribosomes do not accumulate (constraints rRNA and cap R in Table 1 are limiting). Light green diamonds are experimental data from Bremer and Dennis[20], black triangles are the data from Bremer and Dennis[20] corrected for rRNA degradation[28]. Data was converted to mmol g$^{-1}$ with *E. coli* dry masses from Bremer and Dennis[20] and the number of nucleotides in rRNA ($n_{rRNA}$) at $x_{rP} = 36\%$.

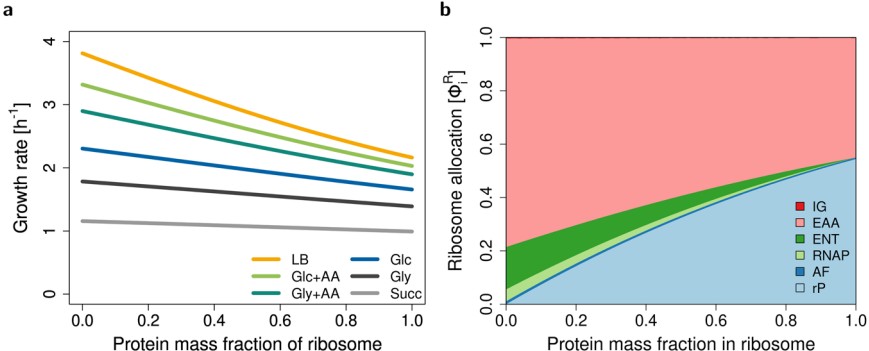

**Fig. 3 Maximum growth rate and ribosome allocations as functions of ribosomal protein fraction $x_{rP}$ for the base model. a** Maximum growth rate for *E. coli* in six different conditions (see Supplementary Table 1). **b** Ribosome allocations $\phi_i^R$ as defined in Eqn. (3), for glucose minimal medium (Glc).

synthesis, and constraints (5) and (7) constitutes the extended RBA model.

Taking rRNA degradation into account leads to maximum growth rates at mixed (RNA+protein) ribosome compositions (Fig. 4). As it turns out, the assumption of a constant $k^{deg}$ in constraint (5) leads to a very shallow optimum (Fig. 4a). To account for the stabilizing influence of rPs on the folded structure, we introduce the non-linear (Hill-type) degradation term (Equation (6) with half-saturation $K = 0.2$ and Hill-factors $n = 2$ or $n = 6$), leading to a pronounced optimum, see Fig. 4b and c.

In the following, we investigate how the optimal ribosome composition depends on growth conditions.

First, we study growth on glucose minimal medium and adjust $k_{max}^{deg}$ such that the optimal ribosome composition matches the experimentally observed value of $x_{rP} = 0.36$ for *E. coli*. We validate the model for the three types of degradation, and we correctly predict the linear dependence of the RNA to protein ratio on growth rate (Supplementary Fig. 3). However, RNAP flux predictions are only realistic when assuming strong cooperativity ($n = 6$). For the other two cases, rRNA degradation in the optimum is too high which leads to overestimated RNAP fluxes (Supplementary Fig. 4).

Second, we predict maximum growth rate as a function of the ribosomal protein fraction in six different growth media. We find that the more proteins cooperate, the less the optimal ribosome composition depends on the growth conditions, see Fig. 4a–c.

Using variable or constant $k_R^{el}$ has no impact on the predicted optimal ribosome composition. As in the base model, variable $k_R^{el}$

leads to predicted non-zero offset of RNA/protein ratio at zero growth rate (Supplementary Fig. 5).

Third, to further understand these results, we plot ribosome allocations for glucose minimal medium, see Fig. 4d–f. Interestingly, at low $x_{rP}$, a large fraction of ribosomes is allocated to the production of RNAP, whereas with increasing $x_{rP}$, this ribosome allocation rapidly drops. In the case of the highest cooperativity, allocations at the optimal $x_{rP}$ are comparable to the base RBA model (without RNA degradation), compare Fig. 4f with Fig. 3b.

Finally, we qualitatively predict that the fraction of degraded rRNA decreases with growth rate (Fig. 5), which is in agreement with experimental observations[28]. This effect gets stronger (and closer to experimental data) with higher rP cooperativity. The quantitative disagreement between the experimental and predicted values is probably due to the simplicity of our model. For example, it does not include other types of RNA or regulatory processes, both of which influence RNAP activity. If we consider RNAP allocation to rRNA ($\bar{k}_{RNAP}^{el} = k_{RNAP}^{el} f_{RNAP}^{act} \phi_{rRNA}^{RNAP}$, where $\phi_{rRNA}^{RNAP}$ is the fraction of RNAP allocated to the synthesis of rRNA), the results get closer to the experimental data (Supplementary Fig. 6).

Based on these results, we conclude that accounting for RNA degradation and cooperative binding of rP can explain the mixed ribosome composition.

**Extreme conditions increase the optimal protein fraction in (archaeal) ribosomes.** As a straightforward extension, we explore

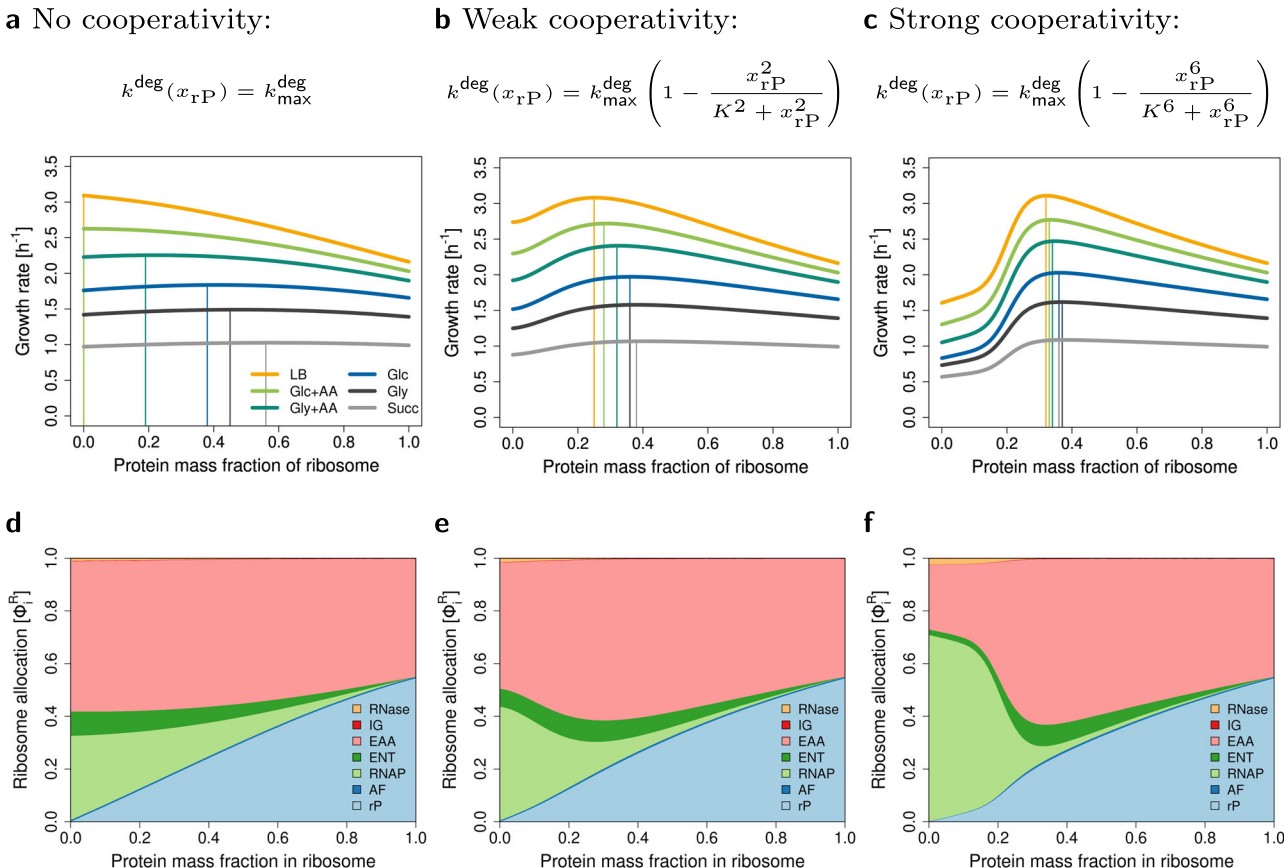

**Fig. 4 Extended model. Accounting for RNA degradation leads to a mixed (RNA+protein) ribosome composition. a–c** Maximum growth rate of *E. coli* in six different conditions (see Supplementary Table 1) for three types of rRNA degradation. **a** No cooperativity of rPs. **b** Weak cooperativity of rPs. **c** Strong cooperativity of rPs. **d–f** Ribosome allocations in glucose minimal medium (Glc) for three types of rRNA degradation. At low protein fractions, rRNA degradation is high, and RNAP (light green) takes up a large amount of cellular resources.

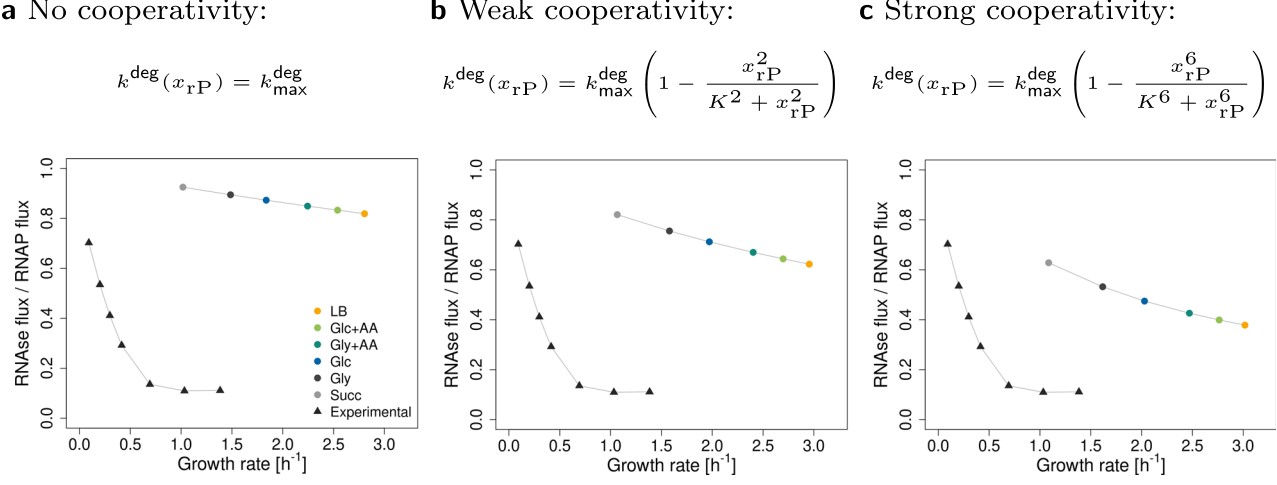

**Fig. 5 Fraction of degraded rRNA at different growth rates.** The extended model recapitulates the experimentally observed decrease in the fraction of degraded RNA with increasing growth rate. The circles are the predicted ratios of RNAse fluxes to RNAP fluxes at different conditions. The triangles represent experimental data from Gausing[28], extracted from the original plot with WebPlotDigitizer[66]. Panels (**a–c**) represent three types of rRNA degradation. **a** No cooperativity of rPs. **b** Weak cooperativity of rPs. **c** Strong cooperativity of rPs.

whether the current model can be adapted to predict the ribosome composition of other organisms. For example, archaeal ribosomes contain 36–50% protein[31], eukaryotic ribosomes 42–50% protein[5,10,11], and mitochondrial ribosomes 51–89% protein[32]. We ask whether this variability can be explained by efficient resource allocation.

It has been hypothesized that the extra archaeal/eukaryotic ribosomal proteins primarily serve to stabilize the ribosomes[33]. This may be particularly important for archaea because they commonly live in extreme conditions, such as high temperatures or low pH, which may lead to higher (misfolding and) degradation of RNA. To mitigate this, archaea might need a higher protein content

**a** No cooperativity:

$$k^{\deg}(x_{\mathrm{rP}}) = k^{\deg}_{\max}$$

**b** Weak cooperativity:

$$k^{\deg}(x_{\mathrm{rP}}) = k^{\deg}_{\max}\left(1 - \frac{x_{\mathrm{rP}}^2}{K^2 + x_{\mathrm{rP}}^2}\right)$$

**c** Strong cooperativity:

$$k^{\deg}(x_{\mathrm{rP}}) = k^{\deg}_{\max}\left(1 - \frac{x_{\mathrm{rP}}^6}{K^6 + x_{\mathrm{rP}}^6}\right)$$

**Fig. 6 The model can be adjusted to predict archaeal protein-rich ribosome composition.** The model was adapted to archaea by increasing $k^\circ_{\max}$ two-fold. The remaining parameters were either kept the same as in *E. coli* (red solid line), or parameters from *Thermococcus* (molecular masses of R and RNAP, transcription and translation rates, see Supplementary Table 1) were used (red dashed line). Panels (**a**–**c**) represent three types of rRNA degradation. **a** No cooperativity of rPs. **b** Weak cooperativity of rPs. **c** Strong cooperativity of rPs.

compared to bacteria. It has been shown that the initial steps in ribosome assembly of the thermophilic archaeon *Sufolobus solfataricus* do not require high temperature and likely involve core proteins that are also present in bacteria. However, completing the assembly requires high temperature, suggesting that these proteins have evolved to cope with such extreme conditions[34,35].

We model this process by increasing $k^{\deg}_{\max}$ which leads to a higher predicted protein content of the ribosome (Fig. 6). Similarly to *E. coli*, the higher the cooperativity, the lower the sensitivity of the optimum to the other parameters. Moreover, when using parameters from *Thermococcus* (see Supplementary Table 1) we observe an increase in ribosomal protein content, in accordance with experimental evidence[31], and predict a decrease in growth rate.

## Discussion

The ribosome is a central player in cellular self-fabrication, placing an upper bound on growth rate. To grow faster, a cell needs more ribosomes which, in turn, requires even more ribosomes to produce themselves. While most catalysts and molecular machines within a cell are proteins, ribosomes stand out by having a substantial (mass) fraction of rRNA, playing a catalytic role. The mass fraction of rPs varies across kingdoms, ranging from approximately 36% in prokaryotes[9] to around 50% in eukaryotes[10], and even higher in eukaryotic mitochondria, reaching up to 89% in *Trypanosoma brucei*[32,36]. This prompts the question: what factors determine the ratio of RNA to protein in ribosomes?

The analysis of our base model (without RNA degradation) suggests that RNA-only ribosomes maximize growth rate (Fig. 3). This results from the lower cost of rRNA synthesis compared to rP synthesis. It remains true even when one accounts for the synthesis of inactive RNAP and enzymes required for nucleotide synthesis[5], which suggests that the costs of rRNA synthesis and associated processes are underestimated in the base model.

In order to explain a mixed (RNA+protein) ribosome, we consider rRNA degradation in our extended model, thereby increasing the costs for RNA synthesis. While rRNA that is already integrated into a ribosome is stable, nascent RNA may be susceptible to degradation[27]. Indeed, it has been experimentally observed that even at maximum growth rate, 10% of newly synthesized rRNA is degraded[28], and the degradation rate increases if ribosome assembly is delayed[27]. Furthermore, when

rRNA is overexpressed in excess of rPs, it is rapidly degraded[25]. Due to the extremely high rates at which rRNA is synthesized, errors become inevitable, necessitating the action of quality control enzymes such as polynucleotide phosphorylase (PNPase) and RNase R to ensure ribosome integrity[37]. The absence of the RNases results in the accumulation of rRNA fragments, ultimately leading to cell death[27,38]. In contrast, protein turnover is negligible[20], and most ribosomal proteins can exist without rRNA and can be reused[39,40]. Therefore, we do not consider protein degradation in our model.

In our resource balance approach, decreasing the RNA content of the ribosome saves resources by reducing RNA turnover. At the same time, protein synthesis costs increase, leading to a mixed (RNA+protein) ribosome at maximum growth rate.

We include RNA degradation in two scenarios. (a) RNA is degraded at a rate proportional to its concentration, or (b) RNA degradation rate decreases non-linearly with ribosomal protein content, since proteins cooperatively protect RNA from degradation[29,41,42]. Both versions of the model predict an optimal mixed (RNA+protein) ribosome. However, without considering cooperative protein binding, optimal ribosome compositions depend on growth conditions. Notably, the higher the cooperativity, the closer the predicted RNAP fluxes and the fraction of degraded rRNA are to experimental data. Yet, more experimental data is needed to decide whether ribosome composition in *E. coli* remains truly independent of growth conditions when the bacterium is evolutionarily adapted to a single environment. Based on these results and available experimental evidence for cooperative protein binding[42], we conclude that scenario (b) is more likely.

Our simple model lumps ribosome assembly and RNA degradation and hence allows multiple explanations for the precise mechanism. On the one hand, proteins may stabilize RNA either by blocking the access of RNases to RNA or by preventing misfolding. Intuitively, this could be explained by the fact that RNA molecules are long, and in order to protect them from misfolding and degradation, a certain critical amount of proteins is needed. Folding intermediates can get trapped in misfolded states and are subsequently degraded as a part of quality control. Proteins may help RNA to avoid these kinetic traps[24,29,43,44]. On the other hand, proteins may increase the rate of ribosome assembly and thereby reduce the number of ribosome intermediates (pre-R in Fig. 7). Indeed, it was observed that rRNA can fold to near-native

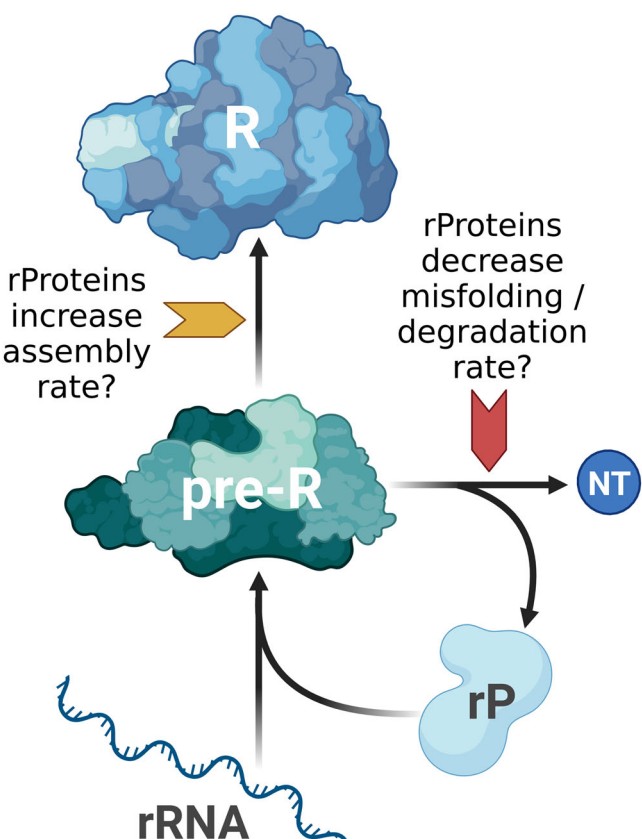

**Fig. 7 Potential mechanisms by which ribosomal proteins affect the biosynthesis of ribosomes.** Ribosomal proteins may decrease the degradation rate or misfolding of ribosome assembly intermediates (pre-R). Alternatively, they may increase the assembly rate. Created with BioRender.com.

conformation[45,46]. Yet, this process is slower than the protein-supported one, especially for long molecules[24,47].

Throughout the manuscript, we make use of two simplifications:

- As in Kostinski and Reuveni[12], we consider ribosomes with different compositions, but equal mass. RNA enzymes, known as ribozymes, are generally smaller than proteins and require only a few nucleotides for catalytic activity[48]. However, such small ribozymes are also inefficient. Increasing their size often improves turnover number, but may impede folding[16,47,49]. Therefore, we consider the case of a large, hard-to-fold, but catalytically efficient RNA-only ribosome.
- We do not consider the effect of protein content on catalytic rates of the ribosomes. Proteins are generally more efficient catalysts than ribozymes[16], yet rRNA is still present in the peptidyl transferase center[17], and translation rate does not increase in ribosomes from different species which have higher protein content[18,19]. Furthermore, despite the modest catalytic rate of peptide bond formation, it does not appear to be the rate-limiting step. Given the size of the substrate molecules (mRNA), diffusion may be the limiting factor[16,50]. Therefore, we assume that enhancing ribosome catalytic rate is not the main reason for the addition of proteins. However, it is possible that proteins stabilize the ribosome structure and thereby indirectly ensure efficient peptide bond formation[16].

In future versions of the model, these assumptions can be relaxed. Furthermore, incorporating other types of RNA (mRNA, tRNA) and energy metabolism, or even constructing a genome-scale

RBA model[51], will likely lead to more quantitative predictions of fluxes and growth rate. A strong indication of this is that including a variable RNAP allocation into the model leads to quantitatively better predictions (see Supplementary Fig. 6). Therefore, in the future, we aim to model RNAP allocation mechanistically. This will involve for example adding other RNA species (mRNA, tRNA), and considering non-specifically bound RNAP which is a considerable fraction of RNAP[52].

In our current model, we approximate the cellular rRNA concentration using the ribosome's rRNA content, see Equation (5). However, rRNA degradation likely begins already during transcription and ribosome assembly. This aspect is not captured in standard RBA as concentrations of all non-catalysts approach zero at maximum growth rates.

In the future, we aim to use growth balance analysis[53,54]. Growth balance analysis allows the integration of nonlinear kinetics, depending not only on catalyst concentrations but also on substrate concentrations. This will enable us to model RNA degradation based on the concentrations of free rRNA or assembly intermediates. While this shift may alter quantitative predictions, such as RNA degradation fluxes and estimates of $k_{max}^{deg}$, the fundamental conclusions drawn from the model are expected to remain unchanged.

To better model protein-rich organisms such as archaea, the model could be expanded by including the temperature dependence of rRNA degradation and assembly in more detail. Apart from $k_{max}^{deg}$, other parameters (e.g. $K$ or $n$ in the Hill function) might change too to capture the effects of extreme conditions. Furthermore, the effects of other extreme conditions (such as pH and osmolarity), and the reasons for the variability of archaeal ribosome composition could also be investigated[55,56]. However, the predictions of our current model are in agreement with the naive expectation that more proteins are required to keep ribosomes stable in harsh conditions. More experimental data is needed to model the archaeal ribosomes realistically.

Additionally, some extremophilic organisms such as the bacteria *Chloroflexus aurantiacus* or *Fervidobacterium islandicum*, exhibit ribosomes with lower protein content (approximately 40%) compared to extremophilic archaea (50%). It has been suggested that protein-rich ribosomes can be traced back to the oldest phylogenetic lineages, with some ribosomal proteins being lost over time[31,57]. Organisms with lower protein content in their ribosomes may have evolved alternative strategies to thrive in extreme conditions. Examples of such strategies include the presence of specific rRNA sequence variants or base modifications, as recently discussed by Nissley et al.[58].

Moreover, certain archaeal species, such as those from Methanobacteriales or Halobacteriales, have transitioned to milder environmental conditions and subsequently shed unnecessary ribosomal proteins[31,57].

To gain a comprehensive understanding of ribosome evolution in response to changing conditions, a thorough phylogenetic analysis is warranted. This analysis should be complemented by measurements of growth rate, translation rate, RNA degradation rate, among other parameters, to delineate the order of protein loss or gain, and the emergence of sequence variations and base modifications.

The protein content of eukaryotic ribosomes in the cytoplasm (approximately 50%) is higher than in bacteria[10]. This is consistent with the lower growth rates seen in eukaryotes like yeast and mammalian cells. Mitochondrial ribosomes show an even higher protein content, ranging from approximately 50% to 89%[32]. This may be advantageous since rP are not made directly in mitochondria but are imported for free from the cytoplasm[59]. Indeed, when we allow a free import of rP in our model, we

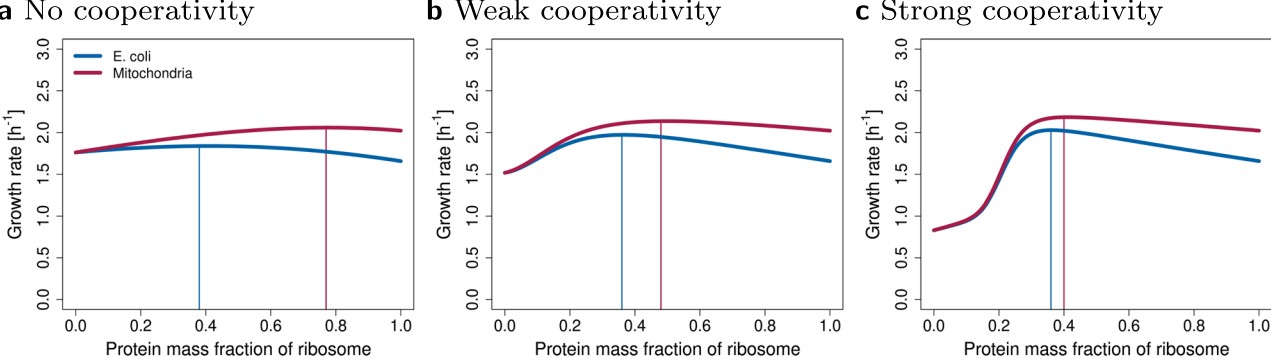

**Fig. 8 The model can be adjusted to predict mitochondrial protein-rich ribosome composition.** All parameters used for the simulation of mitochondria are the same as for *E. coli* in glucose minimal media, except a fraction of rPs can be imported for free from the cytoplasm and does not have to be synthesized. For simplicity, we assumed that 1/3 of rP are imported. (In reality, almost all rP are imported, but mitochondria make additional proteins to provide energy for the whole cell). Panels (**a**–**c**) represent three types of rRNA degradation. **a** No cooperativity of rPs. **b** Weak cooperativity of rPs. **c** Strong cooperativity of rPs.

observe that the optimum moves towards a protein-rich ribosome (Fig. 8). However, to accurately model eukaryotic ribosomes, it is essential to include the synthesis of both cytoplasmic and mitochondrial ribosomes, several different types of RNAPs, transport between nucleus and cytoplasm, and the dynamic interaction between host cells and mitochondria. While the cytoplasm provides ribosomal proteins for mitochondria, mitochondria synthesize enzymes of oxidative phosphorylation and provide ATP back to the host cell.

We hypothesize that eukaryotic cells can afford a higher protein content in their cytoplasmic and mitochondrial ribosomes without affecting the growth rate, and thereby gain additional functionalities that might provide a fitness advantage. Ribosomal proteins participate in translation processes, for example, binding of translation factors, release of tRNA, and translocation. They may also affect the fidelity of translation[60]. Furthermore, they play roles in various cellular processes such as cell proliferation, apoptosis, DNA repair, cell migration, and others[33]. These additional functions might have conferred evolutionary fitness advantages. Nevertheless, the primary role of ribosomal proteins seems to be stabilization and folding of rRNA[33,60].

There are still many open questions about ribosome biogenesis and evolution. Our model could guide future experiments. There are a few studies that assessed the effect of individual rP deletions in *E. coli*, for example, mutation in S10 increased RNA degradation[61], and mutation in L6 led to disrupted ribosomal assembly[62]. A systematic knock-out screen of all ribosomal proteins could be done (as in[63]), complemented with quantification of RNA degradation and misfolding. In case of extremophilic organisms with protein-rich ribosomes, temperature sensitivity could also be assessed. We would expect that deletion of the extra proteins would cause growth defects only at high temperatures. Furthermore, after removal of proteins from archaeal protein-rich ribosomes, laboratory evolution could be performed to see whether growth rate increases beyond wild-type.

Comprehensive datasets, akin to the work of Bremer and Dennis in 2008 for *E. coli*, should be generated for non-standard organisms by measuring various parameters such as transcription and translation rates, ribosome and RNAP activities, and other relevant factors.

Finally, phylogenetic analysis or ribosome evolution across different species and environments could be done.

**Formal comparison with Kostinski and Reuveni (2020).** Our analysis is motivated by the previous work of Kostinski and Reuveni[12], who understand ribosome composition as a competition

between two autocatalytic loops. One loop is responsible for synthesizing rRNA, while the other loop is responsible for rP synthesis, both competing for limited resources. These loops and their constraints, namely, the stoichiometric constraints for rRNA and rP and the capacity constraint for RNAP, are contained in our more detailed RBA model, see Table 1. In addition to these three conditions, Kostinski and Reuveni[12] make two more assumptions: they fix the ribosome allocations $\phi_{rP}^R$ and $\phi_{RNAP}^R$ for the synthesis of rP and RNAP, defined in Eqn. (3).

The resulting upper limits on growth rate can be derived easily by considering the synthesis of rRNA and rP, separately.

- (rP) The stoichiometric constraint for rP is given by $v_{AF} \leq w_{rP}$, see Table 1. Together with the definition of the corresponding ribosome allocation $\phi_{rP}^R = \frac{\mu\, n_{rP}\, w_{rP}}{\bar{k}_R^{el}\, v_{AF}}$, this yields

$$\mu \leq \frac{\bar{k}_R^{el}\, \phi_{rP}^R}{n_{rP}}. \tag{8a}$$

- (rRNA) The stoichiometric constraint for rRNA and the capacity constraint for RNAP are given by $v_{AF} \leq v_{RNAP}$ and $\mu\, n_{rRNA}\, v_{RNAP} \leq \bar{k}_{RNAP}^{el}\, w_{RNAP}$, see Table 1. By multiplication, they imply $\mu\, n_{rRNA}\, v_{AF} \leq \bar{k}_{RNAP}^{el}\, w_{RNAP}$. Together with the definition of the ribosome allocation $\phi_{RNAP}^R = \frac{\mu\, n_{RNAP}\, w_{RNAP}}{\bar{k}_R^{el}\, v_{AF}}$ for the synthesis of RNAP, this yields

$$\mu^2 \leq \frac{\bar{k}_R^{el}\, \bar{k}_{RNAP}^{el}\, \phi_{RNAP}^R}{n_{rRNA}\, n_{RNAP}}. \tag{8b}$$

These upper bounds (8) are Eqns. (2) and (5) in Kostinski and Reuveni[12], after inserting the effective transcription and translation elongation rate constants $\bar{k}_{RNAP}^{el} = k_{RNAP}^{el}\, f_{RNAP}^{act}\, \phi_{rRNA}^{RNAP}$ and $\bar{k}_R^{el} = k_R^{el}\, f_R^{act}$, respectively. Here, $\phi_{rRNA}^{RNAP}$ denotes the fraction of RNAP transcribing rRNA (which we assume to equal one in the rest of this work).

Using Eqn. (4), the two upper bounds (8) can be written as functions of the rP fraction $x_{rP}$, namely as

$$\mu \leq \gamma_{rP}\frac{\phi_{rP}^R}{x_{rP}} \quad \text{and} \quad \mu \leq \gamma_{RNAP}\sqrt{\frac{\phi_{RNAP}^R}{1 - x_{rP}}} \tag{9}$$

with constants $\gamma_{rP}, \gamma_{RNAP} > 0$. For fixed ribosome allocations $\phi_{rP}^R$ and $\phi_{RNAP}^R$, the two curves necessarily intersect at some $0 < x_{rP}^* < 1$, and $\mu(x_{rP}^*)$ is the maximum growth rate allowed by the constraints

---

**Box 1 | Comprehensive models of cellular growth**

Comprehensive models of cellular growth (as used in RBA) need not be genome-scale, but involve explicit synthesis reactions for all catalysts. This is in contrast to traditional metabolic models (as used in FBA) which involve an approximate biomass reaction, thereby fixing biomass composition.

At steady state, the dynamic model of cellular growth yields $\boldsymbol{N}\boldsymbol{v}_{tot} = \mu\,\boldsymbol{c}$ together with the (dry) mass constraint $\boldsymbol{\omega}^T\boldsymbol{c} = 1$. Thereby, $\mu$ is growth rate, $\boldsymbol{v}_{tot}$ is the vector of all fluxes, $\boldsymbol{c}$ is the vector of concentrations, and $\boldsymbol{\omega}$ is the vector of molar masses.

In the constraint-based approach, we consider the (in-)equality system for the fluxes

$$\boldsymbol{N}\boldsymbol{v}_{tot} \geq 0, \quad \boldsymbol{v}_{tot} \geq 0, \quad \text{and} \quad \mu = \boldsymbol{\omega}^T\boldsymbol{N}\boldsymbol{v}_{tot} = \boldsymbol{\omega}^T\boldsymbol{N}_{exc}\boldsymbol{v}_{exc}.$$

Thereby, we assume that all reactions have a given direction, and we use the fact that growth rate is determined by the exchange reactions, cf. Müller et al. (2022)[23].

Finally, concentrations are determined by fluxes via $\boldsymbol{c} = \boldsymbol{N}\boldsymbol{v}_{tot}/\mu$. In particular, concentrations of catalysts are used to formulate additional capacity constraints.

---

considered above, namely the stoichiometric constraints for rP and rRNA, and the RNAP capacity constraint.

Kostinski and Reuveni[12] interpret Eqns. (9) as a trade-off between rRNA and rP production. This effect arises because they fix the ribosome allocations. In particular, Kostinski and Reuveni[12] fix $\phi_{rP}^R$ and $\phi_{RNAP}^R$ to experimental values for *E. coli* (in multiple growth conditions), and find that maximum growth rate occurs close to the current rP fraction ($x_{rP} = 36\%$), and the resulting $\mu(x_{rP})$ is close to the experimental value. If we use their parameters (see Supplementary Table 1), we can exactly reproduce their results (see Supplementary Fig. 7). Our base model provides an explanation for the protein investment costs, giving a proper mechanistic interpretation to the argument presented by Kostinski and Reuveni[12]. Moreover, it is closer to an evolutionary scenario, where a cell can adjust both ribosome composition $x_{rP}$ and ribosome allocations $\phi^R$. However, the base model predicts an optimal ribosome that is RNA-only (for realistic parameters), see Fig. 3a. This is possible because the ribosome allocations are adjusted according to demand. The ribosome allocations corresponding to varying ribosomal protein fraction are illustrated in Fig. 3b. Only the extended model with RNA degradation predicts a mixed (RNA+protein) ribosome at maximum growth rate.

## Methods
Our analysis is based on the small model of a self-replicating cell depicted in Fig. 1 and described below. Constraints are listed in Table 1 and parameters in Supplementary Table 1. For an introduction to resource allocation in next-generation models of cellular growth, including the definition of elementary growth vectors, see Müller et al.[23]. Elementary growth vectors were enumerated using the package efmtool 0.2.0[64] in Python 3.8.13.

Figures 1a and 7 were created with BioRender.com and the remaining figures with R version 4.1.2.

**Model details**. We consider the small model of a self-fabricating cell depicted in Fig. 1 which contains metabolic reactions and macromolecular synthesis reactions. To take into account the limitation of cellular resources, we use three types of capacity constraints: enzyme capacity constraints limit the rate of metabolic reactions, the RNAP capacity constraint limits transcription rate, and the ribosome capacity constraint limits the synthesis rates of all proteins (including the ribosomal proteins).

The cell takes up a carbon source (C) via the reaction

$$r_{IC} : \xrightarrow{IC} C,$$

catalyzed by the importer IC, and forms amino acids (AA),

nucleotides (NT), and ribosomal rRNA (rRNA) via

$$r_{EAA} : n_{AA} \cdot C \xrightarrow{EAA} AA,$$

$$r_{ENT} : n_{NT} \cdot C + AA \xrightarrow{ENT} NT,$$

$$r_{RNAP} : n_{rRNA} \cdot NT \xrightarrow{RNAP} rRNA,$$

catalyzed by the enzymes EAA, ENT, and the RNA polymerase (RNAP). Ultimately, the ribosome R is built from rRNA and ribosomal protein (rP) via

$$r_{AF} : rRNA + rP \xrightarrow{AF} R,$$

catalyzed by the assembly factors AF. The processes above are part of the base model. In an extended model, ribosomal RNA degrades via

$$r_{RNase} : rRNA \xrightarrow{RNase} n_{rRNA} \cdot NT,$$

catalyzed by the RNase. Finally, we consider the synthesis of all proteins (enzymes and ribosomal protein) via the reactions

$$s_i : n_i \cdot AA \xrightarrow{R} i,$$

$$i \in \text{Proteins} = \{IC, EAA, ENT, RNAP, (RNase), AF, rP\},$$

catalyzed by the ribosome.

The resulting stoichiometric matrix and the corresponding flux vector are displayed in Fig. 1b, and parameter values are given in Supplementary Table 1. In fact, the stoichiometric matrix can be partitioned into two submatrices,

$$\boldsymbol{N} = \begin{pmatrix} \boldsymbol{N}_{Met} \\ \boldsymbol{N}_{Cat} \end{pmatrix},$$

corresponding to the metabolites Met = {C, AA, NT, rRNA, rP} and the catalysts Cat = Enz ∪ {R} including the enzymes Enz = {IC, EAA, ENT, RNAP, (RNase), AF} and the ribosome. The (total) flux vector, $\boldsymbol{v}_{tot}$, can be partitioned into two subvectors,

$$\boldsymbol{v}_{tot} = \begin{pmatrix} \boldsymbol{v} \\ \boldsymbol{w} \end{pmatrix},$$

corresponding to the enzymatic reactions $r$ and the protein synthesis reactions $s$.

In general, comprehensive models of cellular growth lead to linear (in-)equality systems for the fluxes, and concentrations are determined by fluxes, see Box 1. In the example, we distinguish enzymatic reactions $r$ and protein synthesis reactions $s$ (with corresponding fluxes $\boldsymbol{v}$ and $\boldsymbol{w}$), and further metabolites Met and catalysts Cat, see above. Explicitly, we study the inequality system

$$\begin{pmatrix} \boldsymbol{N}_{Met} \\ \boldsymbol{N}_{Cat} \end{pmatrix}\begin{pmatrix} \boldsymbol{v} \\ \boldsymbol{w} \end{pmatrix} \geq \boldsymbol{0}, \quad \boldsymbol{v} \geq \boldsymbol{0}, \ \boldsymbol{w} \geq \boldsymbol{0}, \quad \text{and} \quad \mu = \omega_C v_{IC},$$

since $r_{IC}$ is the only exchange reaction.

---

**Box 2 | The determinant method**

(In-)homogeneous linear equality and inequality constraints on a vector $x \in \mathbb{R}^n$ can be summarized by matrices $A' \in \mathbb{R}^{m' \times n}, A'' \in \mathbb{R}^{m'' \times n}$ and vectors $b' \in \mathbb{R}^{m'}, b'' \in \mathbb{R}^{m''}$ as

$$A'x = b', \quad A''x \geq b''.$$

After homogenization, one obtains

$$B'x' = 0, \quad B''x' \geq 0 \quad \text{for} \quad x' = \begin{pmatrix} x \\ 1 \end{pmatrix} \in \mathbb{R}^{n+1},$$

where $B' = (A', -b') \in \mathbb{R}^{m' \times (n+1)}, \quad B'' = (A'', -b'') \in \mathbb{R}^{m'' \times (n+1)}$.

Assume that, for a particular $x$, all inequality constraints are active, that is, $B''x' = 0$. Then,

$$Bx' = 0,$$

where $\quad B = \begin{pmatrix} B' \\ B'' \end{pmatrix} \in \mathbb{R}^{(m'+m'') \times (n+1)}$.

If $B$ is square (that is, if $m' + m'' = n + 1$), then

$$\det B = 0,$$

that is, its determinant is zero.

In the main text, we consider particular (sub-)sets of constraints on the vector of fluxes $v_{\text{tot}}$ in the form $A'v_{\text{tot}} = b', A''v_{\text{tot}} \geq b''$ and assume that, at maximum growth rate, all constraints are active, and the resulting matrix $B$ is square. We compute its determinant, set it to zero, and determine the maximum growth rate from the resulting (quadratic) equation.

---

In fact, only $N_{\text{Met}} \begin{pmatrix} v \\ w \end{pmatrix} \geq 0$ yields non-trivial constraints, since $N_{\text{Cat}} \begin{pmatrix} v \\ w \end{pmatrix} \geq 0$ yields $w_i \geq 0$ for $i \in \text{Enz}$ and $v_{\text{AF}} \geq 0$, already included in $v \geq 0, w \geq 0$. However, $N_{\text{Cat}}$ determines the catalyst concentrations via $\mu c_{\text{Cat}} = N_{\text{Cat}} \begin{pmatrix} v \\ w \end{pmatrix}$ or, explicitly,

$$c_i = w_i/\mu, \quad i \in \text{Enz}, \quad \text{and} \quad c_R = v_{\text{AF}}/\mu. \quad (10)$$

(Recall that the ribosome is formed by the assembly factors).

Now, catalyst concentrations are used to formulate capacity constraints (for importer, metabolic enzymes, and assembly factors),

$$v_i \leq k_i^{\text{cat}} c_i, \quad i \in \{\text{IC}, \text{EAA}, \text{ENT}, \text{AF}\} \subset \text{Enz}, \quad (11a)$$

where $k_i^{\text{cat}}$ are the corresponding enzyme turnover numbers. The capacity constraints for the RNA polymerase, (optionally the RNase), and the ribosome are given by

$$n_{\text{rRNA}} v_{\text{RNAP}} \leq \bar{k}_{\text{RNAP}}^{\text{el}} c_{\text{RNAP}}, \quad (n_{\text{rRNA}} v_{\text{RNase}} \leq k_{\text{RNase}}^{\text{deg}} c_{\text{RNase}}), \quad (11b)$$

and

$$\sum_{i \in \text{Proteins}} n_i w_i \leq \bar{k}_R^{\text{el}} c_R, \quad (11c)$$

respectively. Here, $n_{\text{rRNA}}$ is the number of nucleotides in rRNA, and $n_i$ is the number of amino acids in protein $i$, cf. the stoichiometric coefficients in Fig. 1b. Further, $\bar{k}_{\text{RNAP}}^{\text{el}} = k_{\text{RNAP}}^{\text{el}} f_{\text{RNAP}}^{\text{act}}$ and $\bar{k}_R^{\text{el}} = k_R^{\text{el}} f_R^{\text{act}}$ are the effective transcription and translation elongation rate constants, respectively, and $k_{\text{RNase}}^{\text{deg}}$ is the RNA degradation rate constant.

Finally, catalyst concentrations are expressed by corresponding fluxes in all capacity constraints (11) via Eqns. (10). The stoichiometric, capacity, and (dry) mass constraints described so far are summarized in Table 1, and the parameter values are given in Supplementary Table 1.

In particular, after using (10), the ribosome capacity constraint (11c) takes the form

$$\sum_{i \in \text{Proteins}} \mu n_i w_i \leq \bar{k}_R^{\text{el}} v_{\text{AF}},$$

which suggests the definition of ribosome allocations (ribosome fractions translating certain proteins),

$$\phi_i^{\text{R}} = \frac{\mu n_i w_i}{\bar{k}_R^{\text{el}} v_{\text{AF}}}, \quad i \in \text{Proteins}.$$

Obviously, $\sum_{i \in \text{Proteins}} \phi_i^{\text{R}} \leq 1$. Instead of varying the protein synthesis fluxes $w$, one may vary $v_{\text{AF}}$ (the ribosome synthesis flux) and the (vector of) ribosome allocations $\phi^{\text{R}}$.

Throughout this work, we consider a fixed ribosome mass, but variable ribosomal RNA and protein content,

$$\omega_{\text{R}} = n_{\text{rRNA}} \omega_{\text{NT}} + n_{\text{rP}} \omega_{\text{AA}},$$

where $n_{\text{rRNA}}$ and $n_{\text{rP}}$ are the numbers of nucleotides and amino acids in rRNA and rP, respectively. We define the ribosomal protein (mass) fraction

$$x_{\text{rP}} = n_{\text{rP}} \frac{\omega_{\text{AA}}}{\omega_{\text{R}}},$$

and express $n_{\text{rRNA}}$ and $n_{\text{rP}}$ by $x_{\text{rP}}$,

$$n_{\text{rRNA}} = (1 - x_{\text{rP}}) \frac{\omega_{\text{R}}}{\omega_{\text{NT}}} \quad \text{and} \quad n_{\text{rP}} = x_{\text{rP}} \frac{\omega_{\text{R}}}{\omega_{\text{AA}}}.$$

For variable ribosomal protein fraction $x_{\text{rP}}$ (from 0 to 100%), we maximize growth rate (by varying fluxes under the given constraints).

**Converting constraints from equations/inequalities into matrix form**. For convenience, we describe how to obtain the constraints in Table 1 from the equations/inequalities given in the subsection Model details above.

The capacity constraints for enzymes have the form

$$v_i \leq k_i^{\text{cat}} c_i, \quad i \in \{\text{IC}, \text{EAA}, \text{ENT}, \text{AF}\}.$$

Concentrations $c_i$ can be expressed by synthesis fluxes $w_i$ via

$c_i = w_i/\mu$, and hence

$$v_i \le k_i^{\text{cat}} w_i/\mu.$$

Multiplying by $\mu$ and bringing all terms to one side, we get

$$-\mu v_i + k_i^{\text{cat}} w_i \ge 0$$

or, in vector form,

$$\left(-\mu \; k_i^{\text{cat}}\right)\begin{pmatrix} v_i \\ w_i \end{pmatrix} \ge 0.$$

The row vector is the non-zero part of the row in Table 1 that specifies the capacity constraint cap $i$, for $i \in \{\text{IC}, \text{EAA}, \text{ENT}, \text{AF}\}$.

For the RNAP and RNase capacity constraints (cap RNAP and cap RNase, respectively), we further have $k_i^{\text{cat}} = \bar{k}_{\text{RNAP}}^{\text{el}}/n_{\text{rRNA}}$ and $k_i^{\text{cat}} = k_{\text{RNase}}^{\text{deg}}/n_{\text{rRNA}}$. Hence,

$$\left(-\mu n_{\text{rRNA}} \; \bar{k}_{\text{RNAP}}^{\text{el}}\right)\begin{pmatrix} v_{\text{RNAP}} \\ w_{\text{RNAP}} \end{pmatrix} \ge 0$$

and

$$\left(-\mu n_{\text{rRNA}} \; k_{\text{RNase}}^{\text{deg}}\right)\begin{pmatrix} v_{\text{RNase}} \\ w_{\text{RNase}} \end{pmatrix} \ge 0.$$

Finally, for the ribosome capacity constraint, $\sum_{i \in \text{Proteins}} n_i w_i \le \bar{k}_{\text{R}}^{\text{el}} c_{\text{R}}$, ribosome concentration $c_{\text{R}}$ can be expressed by assembly flux $v_{\text{AF}}$ via $c_{\text{R}} = v_{\text{AF}}/\mu$. Ultimately,

$$\bar{k}_{\text{R}}^{\text{el}} v_{\text{AF}} - \sum_{i \in \text{Proteins}} \mu n_i w_i \ge 0.$$

After expanding the sum, this yields the row in Table 1 that specifies cap R.

Minimum degradation rate is enforced by

$$v_{\text{RNase}} \ge k^{\text{deg}}(1 - x_{\text{rP}}) c_{\text{R}}.$$

As before, ribosome concentration is expressed by assembly flux,

$$v_{\text{RNase}} \ge k^{\text{deg}}(1 - x_{\text{rP}}) v_{\text{AF}}/\mu,$$

which can be written as in Table 1,

$$\left(\mu \; -k^{\text{deg}}(1 - x_{\text{rP}})\right)\begin{pmatrix} v_{\text{RNase}} \\ v_{\text{AF}} \end{pmatrix} \ge 0.$$

As stated in Box 2, growth rate is determined by the exchange fluxes. Indeed, in the dry mass constraint $\boldsymbol{\omega}^T \boldsymbol{c} = 1$, concentrations $\boldsymbol{c}$ can be expressed by fluxes $\boldsymbol{v}_{\text{tot}}$ via $N\boldsymbol{v}_{\text{tot}} = \mu\boldsymbol{c}$ and hence $\boldsymbol{\omega}^T N\boldsymbol{v}_{\text{tot}} = \mu$. Since internal reactions are mass-balanced, only exchange reactions contribute to growth,

$$\boldsymbol{\omega}^T N_{\text{exc}} \boldsymbol{v}_{\text{exc}} = \mu.$$

In our small model, the only exchange reaction is carbon import and hence

$$\omega_{\text{C}} v_{\text{IC}} = \mu.$$

**Symbolic analysis of growth rate maximization.** In order to confirm our numerical results, we also perform a symbolic analysis of growth rate maximization.

The base model involves five stoichiometric constraints (for the species C, AA, NT, RNAP, rP), six capacity constraints (for the reactions catalyzed by IC, EAA, ENT, RNAP, AF, R), and one (dry) mass constraint, cf. Table 1 (without the columns and rows indicated in the caption). They define a linear equality and inequality system with 12 constraints (either $\ge$ or $=$) for 11 fluxes and 1 right-hand side.

We apply the determinant method introduced in Box 2 to the resulting matrix $B \in \mathbb{R}^{12 \times 12}$, and we find

$$\begin{aligned}
0 = \det B \\
\sim \left(\left[\frac{n_{\text{IC}}}{k_{\text{IC}}^{\text{cat}}}\frac{\omega_{\text{NT}}}{\omega_{\text{G}}} + \frac{n_{\text{EAA}}}{k_{\text{EAA}}^{\text{cat}}} + \frac{n_{\text{ENT}}}{k_{\text{ENT}}^{\text{cat}}} + \frac{n_{\text{RNAP}}}{\bar{k}_{\text{RNAP}}^{\text{el}}}\right]n_{\text{rRNA}} + \frac{n_{\text{AF}}}{k_{\text{AF}}^{\text{cat}}}\right)\mu^2 \\
+ \left(\bar{k}_{\text{R}}^{\text{el}}\left[\frac{n_{\text{IC}}}{k_{\text{IC}}^{\text{cat}}}\frac{\omega_{\text{AA}}}{\omega_{\text{G}}} + \frac{n_{\text{EAA}}}{k_{\text{EAA}}^{\text{cat}}}\right] + n_{\text{rP}}\right)\mu - \bar{k}_{\text{R}}^{\text{el}}.
\end{aligned}$$

Using ribosomal protein fraction and rescaling time,

$$n_{\text{rRNA}} = (1 - x_{\text{rP}})\frac{\omega_{\text{R}}}{\omega_{\text{NT}}}, \quad n_{\text{rP}} = x_{\text{rP}}\frac{\omega_{\text{R}}}{\omega_{\text{AA}}}, \quad \text{and} \quad \hat{\mu} = \frac{\mu}{\bar{k}_{\text{R}}^{\text{el}}}\frac{\omega_{\text{R}}}{\omega_{\text{AA}}},$$

we obtain a quadratic equation for maximum growth rate,

$$0 = \left(\alpha + \beta(1 - x_{\text{rP}})\right)\hat{\mu}^2 + \left(\gamma + x_{\text{rP}}\right)\hat{\mu} - 1 \qquad (12)$$

with

$$\alpha = \bar{k}_{\text{R}}^{\text{el}}\frac{n_{\text{AF}}}{k_{\text{AF}}^{\text{cat}}}\left(\frac{\omega_{\text{AA}}}{\omega_{\text{R}}}\right)^2$$

$$\beta = \bar{k}_{\text{R}}^{\text{el}}\left[\frac{n_{\text{IC}}}{k_{\text{IC}}^{\text{cat}}}\frac{\omega_{\text{AA}}}{\omega_{\text{G}}} + \left(\frac{n_{\text{EAA}}}{k_{\text{EAA}}^{\text{cat}}} + \frac{n_{\text{ENT}}}{k_{\text{ENT}}^{\text{cat}}} + \frac{n_{\text{RNAP}}}{\bar{k}_{\text{RNAP}}^{\text{el}}}\right)\frac{\omega_{\text{AA}}}{\omega_{\text{NT}}}\right]\frac{\omega_{\text{AA}}}{\omega_{\text{R}}}$$

$$\gamma = \bar{k}_{\text{R}}^{\text{el}}\left[\frac{n_{\text{IC}}}{k_{\text{IC}}^{\text{cat}}}\frac{\omega_{\text{AA}}}{\omega_{\text{G}}} + \frac{n_{\text{EAA}}}{k_{\text{EAA}}^{\text{cat}}}\right]\frac{\omega_{\text{AA}}}{\omega_{\text{R}}}.$$

For fixed $x_{\text{rP}} \in [0, 1]$, the quadratic equation (12) has one positive solution $\hat{\mu}(x_{\text{rP}})$. To show that it is monotone in $x_{\text{rP}}$, we differentiate (12) and set $\text{d}\hat{\mu}/\text{d}x_{\text{rP}} = 0$. We get

$$0 = -\beta\hat{\mu}^2 + \hat{\mu}$$

which has the positive solution $\hat{\mu} = 1/\beta$. Insertion into (12) yields

$$0 = (\alpha + \beta)\left(\frac{1}{\beta}\right)^2 + \gamma\frac{1}{\beta} - 1 =: \varepsilon,$$

which does not depend on $x_{\text{rP}}$. In fact, if $\varepsilon = 0$, then $\hat{\mu}$ is constant. Otherwise, $\hat{\mu}$ is strictly monotone in $x_{\text{rP}}$ (decreasing if $\varepsilon > 0$ and increasing if $\varepsilon < 0$).

For realistic parameters, $\hat{\mu}$ is decreasing (and $\hat{\mu} < 1/\beta$).

*Approximation.* For realistic parameters, $\alpha \ll \beta \le 1$, and for all $x_{\text{rP}} \in [0, 1]$, we may set $\alpha = 0$ in the quadratic equation (12): For $x_{\text{rP}} \to 0$, obviously $\alpha + (1 - x_{\text{rP}})\beta \to \alpha + \beta \approx \beta$. For $x_{\text{rP}} \to 1$, the crucial quantity $4(\alpha + (1 - x_{\text{rP}})\beta)/(\gamma + x_{\text{rP}})^2 \to 4\alpha/(1 + \gamma)^2 \ll 1$, and the quadratic term can be neglected.

**Numerical growth rate maximization.** We fix growth rate and solve the system of equations (1) and (2) using efmtool 0.2.0[64] in Python 3.8.13. We use bisection search to find the highest growth rate that still enables a feasible solution.

**Reporting summary.** Further information on research design is available in the Nature Portfolio Reporting Summary linked to this article.

## Data availability

All input data and simulation results are available in Zenodo with the identifier https://doi.org/10.5281/zenodo.1045641[65].

## Code availability

All code including the input files and parameters are available in Zenodo with the identifier https://doi.org/10.5281/zenodo.1045641[65]. The calculations were done using efmtool 0.2.0[64] in Python 3.8.13. Figures were generated with R version 4.1.2.

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

## Acknowledgements

This research was funded in part, by the Austrian Science Fund (FWF), Grant https://doi.org/10.55776/P33218.

## Author contributions

Diana Széliová: conceptualization, data curation, formal analysis, investigation, methodology, software, validation, visualization, writing - original draft, writing - review & editing; Stefan Müller: conceptualization, formal analysis, funding acquisition, methodology, validation, writing - original draft, writing - review & editing; Jürgen Zanghellini: conceptualization, formal analysis, funding acquisition, methodology, project administration, resources, supervision, writing - original draft, writing - review & editing.

## Competing interests

The authors declare no competing interests.
