## [Peer Review File · Communications Biology]

Reviewers' comments:

Reviewer #1 (prev. Reviewer #2; Remarks to the Author):

In my opinion, the authors provided a very rigorous response to the different points raised by all reviewers. Overall, this is now a very nice theoretical study exploring the rationalization of RNA to protein ratios in ribosomes based on the shorter stability of RNA. While much more experimental work is needed to probe the role of RNA degradation in setting the RNA to protein ratio - in contrast to for example some more complex relations between the ratio and translation rate - the model gives a compelling simple picture with some predictions being in line with observation, and with specific thoughts about more specific experimental steps to probe the importance of degradation. More generally, I also think that the cross-species comparisons, followed here to explore the properties of ribosomes and their variation, is a very powerful approach which should be taken to heart more commonly when studying fundamental cell physiological properties. I support publication of the manuscript in its current form.

Reviewer #2 (prev. Reviewer #3; Remarks to the Author):

The authors have significantly improved their manuscript. In particular, the authors have added adequate analyses of the effects of growth rate-dependent parameters (e.g., $k_{R^{el}}$, $k_{ENT^{cat}}$, and $\Phi_{rRNA^{RNAP}}$) on the model. I will be happy to recommend its publication if the remaining two comments can be well addressed.

Major:

The modeling of rRNA degradation (previous Major 1). The authors use ribosome concentration as a proxy for free rRNA to model rRNA degradation (eq. 5). Since rRNA degradation is very important for this work, it should be discussed how this approximation might affect the results.

Minor:

The conversion of units (previous Major 5 and 6). Please include the unit conversion in the manuscript. I only learned that the unit of concentrations is mmol per gram of dry mass from the authors' explanation.

Reviewer #3 (Remarks to the Author):

The study "Does ribosome composition maximize growth rate? The role of RNA instability" is very interesting and original, the authors use a mathematical model to analyze the optimal composition of the ribosome in terms of growth rate and energy consumption. The model, taking into account RNA turnover, provide an explanation for the mixed composition of ribosome concluding that increasing the protein to RNA ratio is optimal for protecting RNA from degradation and misfolding.

Comments made by Reviewer n. 1 are in my opinion appropriate.

The first one addresses the fact that ribosomal proteins are involved in roles other than RNA protection. The authors provide an extensive comment on this and in addition as the authors point out stabilization and folding of rRNA are the primary functions, in my opinion, especially if we consider this from an evolutionary point of view.

The second comment addresses the explanation of why ribosomes from different organisms are so different. Also in this case the authors provide extensive comments on this topic proposing for the future a phylogenetic analysis of archaeal evolution to understand the emergence of ribosomes with different composition.

In conclusion I believe that the requests by Referee 1 have received adequate and comprehensive responses and deserves publication

I would like to add an additional comment if possible.

In the Discussion the authors make extensive comments on the composition of bacterial, archaeal and mitochondrial ribosomes while no comments to explain the composition of eukaryotic ribosomes are proposed. I believe adding this type of comment would further improve the quality of the manuscript.

Response to the comments of Reviewer #1 (prev. Reviewer #2)

In my opinion, the authors provided a very rigorous response to the different points raised by all reviewers. Overall, this is now a very nice theoretical study exploring the rationalization of RNA to protein ratios in ribosomes based on the shorter stability of RNA. While much more experimental work is needed to probe the role of RNA degradation in setting the RNA to protein ratio - in contrast to for example some more complex relations between the ratio and translation rate - the model gives a compelling simple picture with some predictions being in line with observation, and with specific thoughts about more specific experimental steps to probe the importance of degradation. More generally, I also think that the cross-species comparisons, followed here to explore the properties of ribosomes and their variation, is a very powerful approach which should be taken to heart more commonly when studying fundamental cell physiological properties. I support publication of the manuscript in its current form.

We would like to thank reviewer #1 for the careful (re-)evaluation of our manuscript and the support for our manuscript. Your feedback is very encouraging.

Response to the comments of Reviewer #2 (prev. Reviewer #3)

We would like to thank reviewer #2 for the careful (re-)evaluation of our manuscript, which we could use to improve its quality significantly. Below we address all comments individually. Comments of the reviewer are represented in blue. The shaded areas indicate additions to the manuscript.

The authors have significantly improved their manuscript. In particular, the authors have added adequate analyses of the effects of growth rate-dependent parameters (e.g., k_R^{el} , k_{ENT}^{cat} , and Φ_{rRNA}^{RNAP}) on the model. I will be happy to recommend its publication if the remaining two comments can be well addressed.

Thank you for an overall positive evaluation.

Major:

The modeling of rRNA degradation (previous Major 1). The authors use ribosome concentration as a proxy for free rRNA to model rRNA degradation (eq. 5). Since rRNA degradation is very important for this work, it should be discussed how this approximation might affect the results.

Thank you for your suggestion. We added the following paragraphs to the discussion section of our manuscript.

In our current model, we approximate the cellular ribosomal ribonucleic acid (rRNA) concentration using the ribosome's rRNA content, see Equation (5). However, rRNA degradation likely begins already during transcription and ribosome assembly. This aspect is not captured in standard resource balance analysis (RBA) as concentrations of all non-catalysts approach zero at maximum growth rates.

In the future, we aim to use growth balance analysis (Dourado et al. 2023; Dourado and Lercher 2020). Growth balance analysis allows the integration of nonlinear kinetics, depending not only on catalyst concentrations but also on substrate concentrations. This will enable us to model ribonucleic acid (RNA) degradation based on the concentrations of free rRNA or assembly intermediates. While this shift may alter quantitative predictions, such as RNA degradation fluxes and estimates of k_{max}^{deg} , the fundamental conclusions drawn from the model are expected to remain unchanged.

Minor:

The conversion of units (previous Major 5 and 6). Please include the unit conversion in the manuscript. I only learned that the unit of concentrations is mmol per gram of dry mass from the authors' explanation.

Thank you for emphasizing the unit conversion concern and offering your suggestion. In response, we've extended Table 2 to comprehensively outline variables and parameters, providing their respective units and concise explanations. We greatly appreciate your insistence—we believe that this addition has significantly enhanced our manuscript's overall readability.

Table 2. Variables and model parameters for *E. coli* in different media, and for *Thermococcus*. If data for *Thermococcus* was not available, we used *E. coli* parameters from glucose minimal medium; LB, Luria-Bertani medium; Glc+AA, glucose + amino acids medium; Gly+AA, glycerol + amino acids medium; Glc, glucose minimal medium; Gly, glycerol minimal medium; Suc, succinate minimal medium. To formulate the constraints in Table 1, the kinetic parameters were converted from s^{-1} , $AA s^{-1}$, and $NT s^{-1}$ to h^{-1} , $AA h^{-1}$, and $NT h^{-1}$, respectively; molar masses were converted to $g mmol^{-1}$; in line with standard practice in constraint-based modeling, concentrations and fluxes are normalized to cell dry mass.

Symbol	Name	LB	Glc+AA	Gly+AA	Glc	Gly	Suc	Thermococcus	Unit	Source
η_{AA}	ω_{AA}/ω_C	0.61	0.61	1.18	0.61	1.18	0.92	0.61	1	
η_{NT}	$(\omega_{NT} - \omega_{AA})/\omega_C$	1.2	1.2	2.34	1.2	2.34	1.82	1.2	1	
η_C		646						646	1	MC [†] CPLX-157
η_{EAA}, η_{ENT}		4875						4875	1	Estimate [‡]
η_{RNAP}		3498						3338	1	(Sutherland and Murakami 2018; Jun et al. 2020)
η_{AF}		3900						3900	1	Estimate [‡]
η_{RNase}		813						813	1	MC [†] EG11259
ω_C	Molar mass carbon source	180	180	92	180	92	118	180	$g mmol^{-1}$	
ω_{AA}	Molar mass amino acid	109						109	$g mmol^{-1}$	BNID [§] 104877
ω_{NT}	Molar mass nucleotide	324.3						324.3	$g mmol^{-1}$	BNID [§] 104886
ω_R	Molar mass ribosome	2300000						3040000	$g mmol^{-1}$	(Kostinski and Reuveni 2020; Acca et al. 1993)
k_{IC}^{cat}	Carbon source import rate	180						180	s^{-1}	BNID [§] 114686
k_{EAA}^{cat}	Enzyme turnover number	10.5	8.5	7	5	3.5	2	5	s^{-1}	Estimate*
k_{ENT}^{cat}	Enzyme turnover number	10.5	8.5	7	5	3.5	2	5	s^{-1}	Estimate*
k_{RNAP}^{el}	Transcription elongation rate	85						25	$NT s^{-1}$	(Bremer and P. Dennis 1996; Gehring and Santangelo 2017)
k_{tr}^{el}	Translation elongation rate	21						8.3	$AA s^{-1}$	(Bremer and P. Dennis 1996), **
k_{AF}^{cat}	Ribosome assembly rate	1/120						1/120	s^{-1}	BNID [§] 102321
k_{RNase}^{deg}	RNase degradation rate	88						88	$NT s^{-1}$	(Fazal et al. 2015)
f_{RNAP}^{tr}	RNAP activity	0.31	0.242	0.188	0.15	0.144	0.132	0.15	1	(Kostinski and Reuveni 2020)
f_R^{tr}	Ribosome activity	0.85						0.85	1	(Hans Bremer and Patrick P Dennis 2008)
k_{RNAP}^{el}	Effective transcription elongation rate $k_{RNAP}^{el} = f_{RNAP}^{tr} k_{RNAP}^{el}$	26.35	20.57	15.98	12.75	12.24	11.22	12.75	$NT s^{-1}$	
k_R^{el}	Effective translation elongation rate $k_R^{el} = f_R^{tr} k_R^{el}$	17.85						17.85	$AA s^{-1}$	
K	Half-saturation constant	0.2						0.2	1	
c	Species concentrations								$mmol g^{-1}$	
v	Metabolic fluxes								$mmol g^{-1} h$	
w	Protein synthesis fluxes								$mmol g^{-1} h$	
μ	Specific growth rate								h^{-1}	
ϕ_i^R	Ribosome allocation to protein i								1	

† MetaCyc ID (Caspi et al. 2018).

‡ estimated from an average protein length of 325 amino acids (BNID 108986) and an approximate number of proteins involved in amino acid/nucleotide synthesis (<https://www.genome.jp/kegg/>), or ribosome assembly (Choi et al. 2020).

§ BioNumbers ID (Milo et al. 2009).

* To consider the nutrient qualities of the different media, we assumed that k_{EAA}^{cat} and k_{ENT}^{cat} are proportional to the experimental growth rates (Suc: 0.4, Gly: 0.7, Glc:1, Gly+AA:1.4, Glc+AA: 1.7, LB: 2.1 h^{-1}). The growth rates were multiplied by 5 so that the maximum k_{EAA}^{cat} corresponds to the average enzyme turnover rate of 10 (Bar-Even et al. 2011).

** An experimentally measured translation rate for *Thermococcus* is unavailable. However, archaeal transcription and translation are likely coordinated, similar to bacteria (French et al. 2007; Proshkin et al. 2010). This suggests an upper bound for the translation rate at approximately $25/3 \approx 8.3 AA s^{-1}$.

Response to the comments of Reviewer #3

We would like to express our gratitude to reviewer #3 for their thorough evaluation, which proved valuable in improving the quality of our manuscript. Below we address all comments individually. Comments of the reviewer are represented in blue. The shaded areas indicate additions to the manuscript.

The study “Does ribosome composition maximize growth rate? The role of RNA instability” is very interesting and original, the authors use a mathematical model to analyze the optimal composition of the ribosome in terms of growth rate and energy consumption. The model, taking into account RNA turnover, provide an explanation for the mixed composition of ribosome concluding that increasing the protein to RNA ratio is optimal for protecting RNA from degradation and misfolding.

Comments made by Reviewer n. 1 are in my opinion appropriate.

The first one addresses the fact that ribosomal proteins are involved in roles other than RNA protection. The authors provide an extensive comment on this and in addition as the authors point out stabilization and folding of rRNA are the primary functions, in my opinion, especially if we consider this from an evolutionary point of view.

The second comment addresses the explanation of why ribosomes from different organisms are so different. Also in this case the authors provide extensive comments on this topic proposing for the future a phylogenetic analysis of archaeal evolution to understand the emergence of ribosomes with different composition.

In conclusion I believe that the requests by Referee 1 have received adequate and comprehensive responses and deserves publication

Thank you very much for your motivating evaluation.

I would like to add an additional comment if possible. In the Discussion the authors make extensive comments on the composition of bacterial, archaeal and mitochondrial ribosomes while no comments to explain the composition of eukaryotic ribosomes are proposed. I believe adding this type of comment would further improve the quality of the manuscript.

Thank you very much for your suggestion, which we've implemented as follows.

The protein content of eukaryotic ribosomes in the cytoplasm (approximately 50 %) is higher than in bacteria (Wilson and Cate 2012). This is consistent with the lower growth rates seen in eukaryotes like yeast and mammalian cells. Mitochondrial ribosomes show an even higher protein content, ranging from approximately 50 % to 89 % (Moore 2019). This may be advantageous since ribosomal protein (rP) are not made directly in mitochondria but are imported “for free” from the cytoplasm (Woellhaf et al. 2014). Indeed, when we allow a “free” import of rP in our model, we observe that the optimum moves towards a protein-rich ribosome (Figure 8). However, to accurately model eukaryotic ribosomes, it is essential to include the synthesis of both cytoplasmic and mitochondrial ribosomes, several different types of RNA polymerases (RNAPs), transport between nucleus and cytoplasm, and the dynamic interaction between host cells and mitochondria. While the cytoplasm provides ribosomal proteins for mitochondria, mitochondria synthesize enzymes of oxidative phosphorylation and provide ATP back to the host cell.

We hypothesize that eukaryotic cells can “afford” a higher protein content in their cytoplasmic and mitochondrial ribosomes without affecting the growth rate, and thereby gain additional functionalities that might provide a fitness advantage. Ribosomal proteins participate in translation processes,

for example, binding of translation factors, release of tRNA, and translocation. They may also affect the fidelity of translation (Nikolay et al. 2015). Furthermore, they play roles in various cellular processes such as cell proliferation, apoptosis, DNA repair, cell migration, and others (Kisly and Tamm 2023). These additional functions might have conferred evolutionary fitness advantages. Nevertheless, the primary role of ribosomal proteins seems to be stabilization and folding of rRNA (Nikolay et al. 2015; Kisly and Tamm 2023).

References

- Acca, Marco, Maurizio Bocchetta, Elena Ceccarelli, Roberta Creti, Karl O. Stetter, and Piero Cammarano. 1993. "Updating Mass and Composition of Archaeal and Bacterial Ribosomes. Archaeal-like Features of Ribosomes from the Deep-Branching Bacterium *Aquifex pyrophilus*" [in en]. *Systematic and Applied Microbiology* 16 (4): 629–637.
- Bar-Even, Arren, Elad Noor, Yonatan Savir, Wolfram Liebermeister, Dan Davidi, Dan S. Tawfik, and Ron Milo. 2011. "The Moderately Efficient Enzyme: Evolutionary and Physicochemical Trends Shaping Enzyme Parameters." PMID: 21506553, *Biochemistry* 50 (21): 4402–4410.
- Bremer, H, and PP Dennis. 1996. "Modulation of chemical composition and other parameters of the cell by growth rate. *Escherichia coli* and *Salmonella*: cellular and molecular biology." *American Society for Microbiology*, 1553–1568.
- Bremer, Hans, and Patrick P Dennis. 2008. "Modulation of chemical composition and other parameters of the cell at different exponential growth rates." *EcoSal Plus* 3 (1): 10–1128.
- Caspi, Ron, Richard Billington, Carol A Fulcher, Ingrid M Keseler, Anamika Kothari, Markus Krumnacker, Mario Latendresse, Peter E Midford, Quang Ong, Wai Kit Ong, et al. 2018. "The MetaCyc database of metabolic pathways and enzymes." *Nucleic acids research* 46 (D1): D633–D639.
- Choi, Eunsil, Hyerin Jeon, Jeong-Il Oh, and Jihwan Hwang. 2020. "Overexpressed L20 rescues 50S ribosomal subunit assembly defects of *bipA*-deletion in *Escherichia coli*." *Frontiers in microbiology* 10:2982.
- Dourado, Hugo, and Martin J Lercher. 2020. "An analytical theory of balanced cellular growth." *Nature communications* 11 (1): 1226.
- Dourado, Hugo, Wolfram Liebermeister, Oliver Ebenhöf, and Martin J Lercher. 2023. "Mathematical properties of optimal fluxes in cellular reaction networks at balanced growth." *PLOS Computational Biology* 19 (6): e1011156.
- Fazal, Furqan M, Daniel J Koslover, Ben F Luisi, and Steven M Block. 2015. "Direct observation of processive exonuclease motion using optical tweezers." *Proceedings of the National Academy of Sciences* 112 (49): 15101–15106.
- French, S. L., T. J. Santangelo, A. L. Beyer, and J. N. Reeve. 2007. "Transcription and Translation are Coupled in Archaea" [in en]. *Molecular Biology and Evolution* 24 (4): 893–895.
- Gehring, Alexandra M., and Thomas J. Santangelo. 2017. "Archaeal RNA polymerase arrests transcription at DNA lesions." *Transcription* 8 (5): 288–296.
- Jun, Sung-Hoon, Jaekyung Hyun, Jeong Seok Cha, Hoyoung Kim, Michael S Bartlett, Hyun-Soo Cho, and Katsuhiko S Murakami. 2020. "Direct binding of TFE α opens DNA binding cleft of RNA polymerase." *Nature communications* 11 (1): 1–12.
- Kisly, Ivan, and Tiina Tamm. 2023. "Archaea/eukaryote-specific ribosomal proteins-guardians of a complex structure." *Computational and Structural Biotechnology Journal*.
- Kostinski, Sarah, and Shlomi Reuveni. 2020. "Ribosome Composition Maximizes Cellular Growth Rates in *E. coli*." *Physical Review Letters* 125 (2): 028103.

- Milo, Ron, Paul Jorgensen, Uri Moran, Griffin Weber, and Michael Springer. 2009. "BioNumbers—the database of key numbers in molecular and cell biology." *Nucleic Acids Research* 38 (suppl_1): D750–D753.
- Moore, Peter B. 2019. "In Which the Deity Attempts To Make a Ribose-Free Ribosome." *Biochemistry* 58 (6): 431–432.
- Nikolay, Rainer, David van den Bruck, John Achenbach, and Knud H Nierhaus. 2015. "Ribosomal proteins: role in ribosomal functions." *eLS*, 1–12.
- Proshkin, Sergey, A. Rachid Rahmouni, Alexander Mironov, and Evgeny Nudler. 2010. "Cooperation Between Translating Ribosomes and RNA Polymerase in Transcription Elongation" [in en]. *Science* 328 (5977): 504–508.
- Sutherland, Catherine, and Katsuhiko S Murakami. 2018. "An introduction to the structure and function of the catalytic core enzyme of Escherichia coli RNA polymerase." *EcoSal Plus* 8 (1).
- Wilson, Daniel N., and Jamie H. Doudna Cate. 2012. "The Structure and Function of the Eukaryotic Ribosome" [in en]. *Cold Spring Harbor Perspectives in Biology* 4 (5): a011536.
- Woellhaf, Michael W, Katja G Hansen, Christoph Garth, and Johannes M Herrmann. 2014. "Import of ribosomal proteins into yeast mitochondria." *Biochemistry and Cell Biology* 92 (6): 489–498.